# Cationic Materials for Gene Therapy: A Look Back to the Birth and Development of 2,2-Bis-(hydroxymethyl)Propanoic Acid-Based Dendrimer Scaffolds

**DOI:** 10.3390/ijms242116006

**Published:** 2023-11-06

**Authors:** Silvana Alfei

**Affiliations:** Department of Pharmacy, University of Genoa, Viale Cembrano 4, 16148 Genova, Italy; alfei@difar.unige.it; Tel.: +39-010-3532692

**Keywords:** gene therapy, cell transfection, non-viral carriers, cationic dendrimers, 2,2-bis(hydroxymethyl)propanoic acid, amino acids-modified dendrimers

## Abstract

Gene therapy is extensively studied as a realistic and promising therapeutic approach for treating inherited and acquired diseases by repairing defective genes through introducing (transfection) the “healthy” genetic material in the diseased cells. To succeed, the proper DNA or RNA fragments need efficient vectors, and viruses are endowed with excellent transfection efficiency and have been extensively exploited. Due to several drawbacks related to their use, nonviral cationic materials, including lipidic, polymeric, and dendrimer vectors capable of electrostatically interacting with anionic phosphate groups of genetic material, represent appealing alternative options to viral carriers. Particularly, dendrimers are highly branched, nanosized synthetic polymers characterized by a globular structure, low polydispersity index, presence of internal cavities, and a large number of peripheral functional groups exploitable to bind cationic moieties. Dendrimers are successful in several biomedical applications and are currently extensively studied for nonviral gene delivery. Among dendrimers, those derived by 2,2-bis(hydroxymethyl)propanoic acid (b-HMPA), having, unlike PAMAMs, a neutral polyester-based scaffold, could be particularly good-looking due to their degradability in vivo. Here, an overview of gene therapy, its objectives and challenges, and the main cationic materials studied for transporting and delivering genetic materials have been reported. Subsequently, due to their high potential for application in vivo, we have focused on the biodegradable dendrimer scaffolds, telling the history of the birth and development of b-HMPA-derived dendrimers. Finally, thanks to a personal experience in the synthesis of b-HMPA-based dendrimers, our contribution to this field has been described. In particular, we have enriched this work by reporting about the b-HMPA-based derivatives peripherally functionalized with amino acids prepared by us in recent years, thus rendering this paper original and different from the existing reviews.

## 1. Introduction

Dendrimers are synthetic polymers that differ from traditional polymers because radially symmetric macromolecules, characterized by a highly tree-like branched globular structure, low polydispersity index, the presence of internal cavities, and the occurrence of a large number of functional groups on the periphery [1]. Dendrimers are spherical polymers with a core, branches, and terminal groups. A dendrimer’s generation (G) refers to the number of times the branches are repeated. A higher generation means more branches and terminal groups [2]. These characteristics have ensured that dendrimers have been successful in various biomedical applications, such as drug delivery, biosensors, diagnostic imaging, gene delivery, etc. [3].

A particular application of dendrimers in great development regards gene therapy, which is a therapeutic approach aiming at repairing defective genes through the introduction (transfection) in the diseased cells of the “healthy” genetic material [4]. To be successful, genetic materials need to be transported and delivered by proper vectors, which could be viral or nonviral [5]. Viral vectors are endowed with high transfection efficiency (TE), but several drawbacks, such as potential toxicity and immunogenicity, still limit their use [5,6], thus encouraging incessant research for designing and developing other new vectors of a non-viral type. Synthetic nonviral carriers include positively charged materials such as cationic liposomes, lipidic nanoparticles, polymer vectors, and cationic dendrimers [6].

In particular, dendrimers that have nitrogen atoms that can be protonated at physiological pH have aroused the interest of researchers; this is because polycationic systems, being able to establish electrostatic interactions with the phosphate anionic groups of DNA or RNA fragments, can promote their compaction, thus forming nanoparticles (dendriplexes), whose polymeric part protects the genetic material from degradation by endonucleases during its route to the cell nucleus [4].

The nanosized complexes made of nucleic acids and dendrimers succeed in entering into the cells by numerous endocytic mechanisms [7]. In this context, poly(amidoamine) (PAMAM) dendrimers are a well-known and commercially available class of cationic dendrimers, which have been explored as vectors of genetic materials since they possess terminal amino groups that can interact with phosphate groups of nucleic acids. Additionally, their high-density tertiary amino groups scattered throughout their scaffold are responsible for the phenomenon known as “proton sponge”, which allows the dendriplex to escape the endosome, thus avoiding lysosomal degradation [7,8]. Several studies have been reported about the use of PAMAM dendrimers as gene delivery systems. Among PAMAMs, those of second generation (G2) have shown higher transfection efficiency (TE) and lower cytotoxicity than G4 and G5 PAMAMs in various cell lines [9]. Moreover, G2 PAMAMs can be easily modified with different functional groups to enhance their biocompatibility and targeting ability [10]. One study reported that G2 PAMAM conjugated with arginine and fluorinated moieties exhibited improved gene TE and reduced cytotoxicity compared to unmodified dendrimers [9]. Additionally, it was demonstrated that G2 PAMAMs decorated with alkyl sulfonyl hydrophobic tails were capable of forming stable complexes with plasmid DNA (pDNA) and showed improved gene expression both in vitro and in vivo [10]. Among cationic dendrimers, poly(lysine) (PLL) and materials that integrate lysine in their structure were promising drug delivery systems [11]. Like PAMAMs, PLLs are cationic macromolecules capable of electrostatically interacting with both genetic material and cell membranes, thus being efficiently internalized and enabling intracellular gene delivery [11]. Although PLL is intrinsically pH-responsive, the conjugation with different functional groups could impart smart, stimuli-responsive traits [11]. Recently, Gorzkiewicz et al. have synthesized two new lysine-based dendrimers (D3K2 and D3G2) and have investigated their transfection potential in two cell line models. Cationic D3K2 dendrimer demonstrated to possess high TE, thus enabling the intracellular accumulation of large nucleic acid molecules like plasmids [12]. Unfortunately, an important problem of polycationic systems such as PAMAMs is their cytotoxicity and hemolytic toxicity, which limits their applications in vivo [13].

Unlike PAMAMs, dendrimers derived by 2,2-bis(hydroxymethyl)propanoic acid (b-HMPA) and having a polyester-based scaffold could be particularly good-looking due to their degradability in vivo [14]. In this regard, the objective of this paper was to drive the attention of readers toward this unique class of biocompatible dendrimers for improving their finalization as gene delivery systems. To this end, with respect to the recent book by Malkoch and García-Gallego reporting on b-HMPA-based dendritic polyesters [15], after a discussion on gene therapy and the main cationic materials proposed for the transport and delivery of genetic materials, the present paper tells the history of the birth and development of the main b-HMPA derived dendrimer scaffolds. The foremost synthetic strategies and the most interesting dendron and dendrimer architectures, which have been obtained by b-HMPA-derived monomers, have also been reviewed. The commercially available b-HMPA-based dendrons and dendrimers that are useful for further functionalization to achieve cationic materials for gene delivery application have been reported in a reader-friendly table. Additionally, our contribution to this field has been described to add novelty to this work. Specifically, we have reported about the b-HMPA-based dendrimer derivatives peripherally functionalized with amino acids we prepared in recent years and the successful synthetic strategies designed and exploited to prepare them.

## 2. Gene Therapy and Non-Viral Carriers

Gene therapy refers to all procedures and protocols aimed at inserting (transfection) a gene, segments of genes, or nucleotides into the cells of a diseased subject because of genetic defects to produce a therapeutic effect [16].

This therapeutic approach has received considerable attention as a potential method for the treatment of serious genetic disorders such as combined immunodeficiency, cystic fibrosis, Parkinson’s disease, and other neurodegenerative diseases. Additionally, gene therapy could be an alternative curative method to the classical chemotherapy used for the treatment of cancer [17]. Both inherited genetic diseases (e.g., hemophilia and sickle cell disease) and acquired disorders (e.g., leukemia) have been treated with gene therapy.

Gene therapy can be applied to somatic cells; therefore, the defect will be treated only in that subject or germinal cells, where the transmission of characters is also ensured to subsequent generations [18]. Since man would be able to modify the genome of future generations, this second approach would involve enormous problems from an ethical point of view.

To have success in gene therapy, it is necessary first to identify the portion of the gene responsible for the pathology. Subsequently, it is essential to select the more proper technique for isolating the fragment of DNA containing the information needed to synthesize the protein required for correcting the pathology. Additionally, it is basic to clone and amplify the correct nucleotide sequence [19]. Figure 1 schematically represents these mandatory steps.

The uptake of free DNA through the permeation of the cytoplasmic membrane is strongly limited due to the large size and negative charge of the nucleic acids. Attempts to introduce free DNA into cells through electroporation, the use of a gene-gun, or the direct injection with micro-syringes into target tissues have yielded limited results [20].

For this reason, research is currently oriented towards transferring the genetic material of interest into the cells through appropriate vectors that can compact, transport, and protect it up to the nucleus, thus preventing rapid degradation by blood nucleases [21]. Figure 2 effectively illustrates the stages of the transfection process.

First, the vector and the genetic material form a complex internalized in the cell through endocytosis. Once in the cell, the complex leaves the endosome, and the genetic material still protected by the vector reaches and enters the nucleus, where it is released by the vector, transcribed, and subsequently translated into the healthy proteins by the nuclear transcriptional machine, while the vector is degraded [22]. A pivotal point in carrier-assisted gene therapy is the identification of the most suitable vectors for the transport of genetic material. A well-trodden route involves the use of viruses, including retroviruses and adenoviruses, as carriers of genetic material due to their efficiency in transporting DNA and RNA in numerous cell lines [23].

However, some problems associated with the use of viral vectors, such as potential toxicity, immunogenicity, and the difficulty of large-scale transfer procedures, still limit their use, thus encouraging incessant research for designing and developing other new vectors of a non-viral type [23].

The major non-viral vectors currently available offer potential routes for DNA compaction and transport, but unlike viral vectors, they exhibit significantly reduced TE. In fact, while viruses possess intrinsic tools for overcoming the host’s cellular barriers and immune defense mechanisms, non-viral vectors are impeded by numerous intracellular obstacles [24]. Nonetheless, biocompatibility and relatively simple large-scale production make some non-viral vectors highly promising materials for gene therapy [25].

Several aspects must be taken into consideration to achieve efficient transfection using non-viral vectors, including the vector’s ability to complex the genetic material and the stability of the resulting complex. Moreover, the formed complex’s size and overall charge density, cellular uptake, ability to escape the endosomal compartment, and aptitude to release the genetic material at the nuclear level without undergoing early degradation processes can strongly affect the TE [26].

The search for non-viral vectors has focused on identifying polycationic compounds capable of forming complexes with DNA or RNA and overcoming the barriers to gene transport in vivo and in vitro [27]. It is known that polycationic systems, through electrostatic interactions with the negative charges of the phosphate groups of the nucleic acids, lead to their overall neutralization and the consequent compaction of the nucleotide fragment [26,27].

The dimension and charge of obtained polyplexes are the two main parameters influencing the TE and the vector-DNA complexes’ absorption type. Furthermore, the concentration of DNA, the pH, the type of buffer, the salt concentration, and the N/P ratio, where N indicates the number of protonable groups of the vector and P the number of phosphate groups of the genetic material, could be pivotal factors conditioning the transfection process [25,28,29].

The size of the complex is crucial for the internalization by endocytosis. Several uptake studies suggested that the optimal size for non-viral vector-DNA complexes is between 70 and 90 nm and that their cellular uptake proceeds via various endocytic pathways, such as clathrin-dependent endocytosis or micropinocytosis [30].

Vectors have been developed that are able to interact with specific receptors and promote the release of transported material, specifically in cells and tissues of interest (targeting); this was possible by inserting particular ligands such as asialoglycoprotein, Epidermal Growth Factor (EGF), folate, integrins, transferrin, and some disaccharides such as lactose and mannose, which were recognized by specific receptors expressed in the desired tissues [31]. The interaction of the ligand with the receptors existing on the cell surface determined an internalization of the vector-DNA complex through more effective clathrin-dependent endocytosis [31].

Another important point for an efficient transfection activity is the DNA-vector complex’s ability to escape the endosome formed upon endocytosis, thus avoiding early lysosomal degradation.

The ATP-mediated accumulation of protons in the endosome makes its and the lysosome cellular compartments much more acidic (pH 5.0–6.2) than the cellular cytoplasm (pH 7.4) [1]. The acidic environment of endosomes can be exploited by the non-viral vectors complexing DNA to escape cytoplasmic degradation [7].

Various macromolecular systems endowed with amino groups with pKa values close to the lysosomal one have been shown to exhibit a “proton sponge” effect [7]. When complexed with DNA and incorporated into the cell, these systems are capable of buffering the endosomal vesicle, causing an increase in the ATP-mediated proton pump, which leads to the entry of chloride ions resulting in swelling and in the lysis of the endosome, thus releasing the vector-DNA complex in the cytoplasm [32].

Once released into the cytoplasm, the vector-DNA complex must overcome additional obstacles on its way to the cell nucleus.

The mobility of a free DNA plasmid in the cytoplasm is negligible, probably due to the elements of the cytoskeleton that prevent the diffusion of a large number of molecules, acting as efficient cellular sieves [33].

On the contrary, due to the interaction with the polycationic systems, the compaction of the DNA favors the movement of the complexed DNA from the cell membrane toward the nucleus [34].

The transition of the polyplex from the cytoplasm into the nucleus (trafficking) occurs by passive diffusion through the Nuclear Pore Complex (NPC) and is possible only for compounds with a diameter between 9 and 11 nm [35].

Dividing cells often exhibit a greater tendency to be transfected than non-mitotic cells, indicating that the DNA can reach the nucleus during the nuclear envelope disruption, which is critical for cell division [36].

Transfection studies performed using DNA plasmids complexed with cationic vectors have shown markedly higher levels of gene expression than that observed using free pDNA, thus suggesting that a positively charged vector is capable of exerting a targeted nuclear effect [37].

Unfortunately, while DNA-cationic carrier systems showed effective gene delivery in vitro, they did not demonstrate similar efficiency in in vivo delivery due to the high instability of the nucleic acids under physiological conditions. Additionally, physiological salt concentration (150 mM) often promotes aggregation of cationic complexes, thus leading to potentially dangerous vascular blockages [38].

Moreover, whey proteins such as albumin quickly bind cationic complexes, thus hindering their cellular uptake and promoting their aggregation, possibly triggering phagocytosis and degradation processes [39]. A successful practice consists of equipping the polyplexes with neutral hydrophilic coatings such as polyethylene glycol (PEG) (PEGylation) to address these issues [40].

In the following Section 2.2.1, Section 2.2.2, Section 2.2.3, Section 2.2.4, Section 2.2.5, Section 2.2.6 and Section 2.2.7, a selection of the main types of non-viral vectors studied so far has been discussed.

### 2.1. Lipidic Carriers

One of the first strategies for introducing exogenous genetic material into cells involved the use of cationic liposomes. Liposomes are artificial lipid vesicles with a positive electrical charge, which are demonstrated to be capable of incorporating DNA. Once enclosed in the liposome, DNA can be internalized in the cell, released into the cytosol, and transported to the nucleus.

The first studies were carried out starting from the mid-1970s, and in 1987, the term lipofection was coined to describe the transfection of genes based on lipid vectors [41].

Nowadays, numerous lipid-based transfection reagents are commercially available, including N-[1-(2,3-dioleyloxy)propyl]-N,N,N-trimethylammonium chloride (DOTMA), 2,3-dioleyloxy-N-[2(carboxamidospermine)ethyl]-N,N-dimethyl-1-propanaminotrifluoroacetate (DOSPA), 1,2-dioleoyl-3-trimethylammoniumpropane (DOTAP), and dioctylamidoglycyl-spermine (DOGS) [42] (Figure 3).

The mechanism of gene transfer through the use of cationic lipoplexes has been thoroughly reviewed, and it has been suggested that lipoplexes are released into the cytoplasm by direct fusion with the cell plasma membrane [43]. The prevailing thesis is that the main mechanism of liposome-mediated gene transfer occurs through endocytosis.

Following cellular uptake, the lipoplexes destabilize the endosomal membrane, and a reorganization of the anionic phospholipids takes place with a mechanism known as flip-flop, in which the DNA fragment bearing the useful information is discharged into the cytoplasm [38].

Cationic lipids are composed of three structural domains: a cationic head, a hydrophobic tail, and a linker between the two [44].

Using different types of reagents for their synthesis, it is possible to vary each of these domains, thus producing liposomes with different characteristics. Structure-activity relationship (SAR) studies on a class of cationic lipopolyamines have highlighted that the density and nature of the cationic head are responsible for the transfection properties of such lipids and that the hydrocarbon moiety can be manipulated without affecting the ability to transfer a gene.

Transfection studies have shown that by replacing the cationic ammonium groups with arsenium or phosphonium ones (Figure 4), phosphono lipids with significantly lower cytotoxicity than the quaternary ammonium analogs can be developed [6]. Findings evidenced that for these systems, the in vitro TE increased as the number of methylene units (n) between the phosphate group and the cationic moiety increased (n = 3 > n = 2 > n = 1).

Chabaud et al. [45] attempted to improve the TE of cationic lipids in vivo by inserting uridine, thus forming a cationic lipid nucleoside (Figure 4). These lipids can interact with DNA base pairs via hydrogen bonding and π-π interactions, in addition to the typical electrostatic interactions between protonated amino groups and anionic phosphate groups of DNA [45].

Several studies suggest that the length and type of aliphatic chain can influence the TE of lipid carriers, and among the many cationic lipids, lipids with multivalent cationic head groups are expected to be potent transfection reagents [46]. Mochizuki et al. prepared calix [4] arene-based lipids with different alkyl chain lengths from C3 to C15 and evaluated the relationship between the alkyl chain length and the TE [46]. Among all lipoplexes developed, C6 lipoplexes exhibited the highest TE. The gene expression with C9 and C12 lipoplexes was slightly lower than that with C6 lipoplexes. C3 lipoplexes hardly induced gene expression, while C15 lipoplexes exhibited no complexation with pDNA [46].

In addition to varying the length and type of aliphatic chain, various hydrophobic moieties have been used to promote transfection.

In this regard, to promote phase transformation, endosome escape, and the release of genetic material, Du et al. created cholesterol-amino-phosphate (CAP) lipids RNA delivery systems by incorporating cholesterol, phospholipids, and amines [47]. More recently, cholesterol [48,49] and other steroids [50] were used instead of aliphatic chains to evaluate the importance of the rigidity of these systems and their biodegradability. Figure 5 shows examples of cholesterol and cholestane derivatives.

The cholesterol derivatives are particularly advantageous as they possess greater stability. Furthermore, the use of amphiphilic cationic bile acids has led to a significant increase in TE in vitro (Figure 6). 

In particular, the combination of amphiphilic deoxycholic acid and cationic PAMAM resulted in macromolecules able to co-deliver doxorubicin and pDNA. Additionally, the as-prepared NPs had low cytotoxicity and could achieve a TE of up to 74% in the 293 T cells [51]. Similarly, the amphiphilic combination of bile acids, such as deoxycholic acid and the cationic polyethylene imine (PEI), efficiently transported the genetic cargo and further enhanced TE of Adenovirus (Ad) in CAR-negative cells [52]. Additionally, the Ad-PEI-bile-acid complex developed by Lee et al. showed enhanced TE and therapeutic efficacy in tumors with low coxsackie and adenovirus receptor expression [53]. Polydiacetylenic derivatives (Figure 6) with protonatable amine or imidazole groups crosslinked using UV light generated interesting structures, such as nanomicelles or nanofibers, which were suitable for plasmid RNA (pRNA) and short interfering RNA (siRNA) delivery [54]. Lipids bearing highly fluorinated alkyl chains have been successfully synthesized and have demonstrated excellent gene transfer success due to their ability to avoid interactions with hydrophilic and lipophilic biocompounds [55].

The main drawback associated with the use of cationic liposomes as carriers for gene delivery is their short plasma half-life. Currently, the most used strategy to create liposomes with a long circulation time consists of covalently bonding hydrophilic PEG polymers, thus creating a polymeric coat that surrounds the surface of the liposome [56]. These PEG-coated liposomes are also known as “stealth liposomes” due to their ability to escape phagocytes of the reticuloendothelial system (RES) or as “sterically stabilized”, as the steric hindrance exerted by the polymer is responsible for the long blood half-life [57]. Precisely, liposome stabilization is due to the presence of highly hydrated PEG hydrophilic groups on their surface, which create a steric barrier preventing interactions with molecular and cellular detrimental components of the organism.

### 2.2. Polymeric Carriers

#### 2.2.1. Poly(L-lysine) (PLL)

Historically, in 1975, Laemmli demonstrated the unique ability of poly(L-lysine) (PLL) to condense DNA [58]. These vectors have been applied for gene transfer in vitro [59] and in vivo [60].

Conventionally, three approaches are applied to synthesize PLL, including solid-phase peptide synthesis (SPPS), ring-opening polymerization (ROP), and chemical-enzymatic synthesis (CES). Generally, different synthesis approaches are chosen to prepare PLLs with different structures, dispersity, and molecular weight [61]. Concerning ROP, the synthesis of PLL proceeds through the conversion of L-lysine protected on the ε amino group to the corresponding cyclic N-carboxy anhydride, which undergoes a ring-opening polymerization induced by a primary amine as an initiator. Protected PLL is then deprotected to free PLL by catalytic hydrogenation (Figure 1).

In general, only PLL structures with a molecular weight > 3 KDa can be effectively condensed with DNA to form stable complexes, thus indicating that the degree of polymerization and the number of primary amino groups are essential to obtain condensation.

At physiological pH, all PLL ε-amino groups are protonated, giving a structure with limited buffer capacity, which, instead, is crucial to facilitate the release of the polyplex from the endosome and avoid degradation.

By introducing histidine residues into the PLL structure and obtaining conjugated acids with a pKa = 6.0, a good buffering capacity can be achieved, which facilitates the release of the polyplex from the endosome [62].

Additionally, the presence of imidazole residues in PLL, such as those of histidine, helps to break down the usually high cytotoxicity of these vectors [62].

In addition to the high toxicity, PLL-DNA complexes tend to form large aggregates and precipitate in relation to the ionic strength of the solution and the N/P ratio [63]. To prevent the formation of insoluble precipitates, the polycation-DNA complexes have been stabilized by surface modification with hydrophilic polymers, including PEG, dextran, and poly(N-[2-hydroxypropyl]methacrylamide) (PHPMA). By this strategy, complexes that have significantly reduced size regardless of the salt concentration of a buffer solution and are resistant to digestion by deoxyribonuclease I (DNase I) were obtained [64].

Various biodegradable conjugated PLLs were prepared to reduce cytotoxicity and promote the release of DNA from the PLL polyplex following endocytosis.

McKenzie et al. successfully condensed DNA with low molecular weight lysine oligomers containing terminal cysteine residues capable of forming complexes via disulfide bridges. The resulting structures showed gene transfer capacity comparable to that of commercially available lipid agents [65].

Park et al. synthesized polymeric structures containing PEG building blocks that were found to protect DNA from nuclease degradation and exhibited zero-order plasmid release kinetics after six weeks [66,67].

The insertion of ester groups in the PLL structures led to hydrolyzable derivatives. Biodegradable nanoparticles (NPs) in the form of monomethoxypoly(ethylene glycol)-poly(lactic-co-glycolic acid)-poly-l-lysine (mPEG-PLGA-PLL) triblock copolymers showed reduced cytotoxicity and significant TE compared to not modified PLLs [68].

Bikram et al. synthesized PLL-PEG multiblock copolymers containing ester moieties and different ratios of histidine residues with enhanced buffering capacity [69]. These compounds showed reduced cytotoxicity and gene TE higher than that of non-derivatized PLLs. Furthermore, it was observed that after three days, the in vivo biodistribution of the intact complexes was detected in the blood circulation, suggesting that PEG chains promoted plasma protein binding, thus masking the polymeric complexes and decreasing their degradation (disopsonization) [69].

Finally, it was reported that poly-D-lysine (DPL) conjugates showed higher transfection capacity than PLL ones, as they were more stable for both lysosomal and peptidase degradation [70].

#### 2.2.2. Polyethyleneimines (PEIs)

Polyethyleneimines (PEIs) are one of the most promising examples of polymers for gene transfection and are considered the gold standard material for this application [71]. There are both branched and linear hetero- and homofunctional PEI structures.

The synthesis of branched PEI (b-PEI) proceeds via the acid-catalyzed polymerization of aziridine (Figure 2A), whereas the synthesis of linear PEI (l-PEI) is achieved via ROP of 2-ethyl-2-oxazolin, followed by acid hydrolysis (Figure 2B) [72].

The TE of PEIs is mainly due to their ability to function as “proton sponges” as they contain different types of protonable nitrogen atoms, conferring a high buffer capacity [72]. In fact, at pH = 7.4 of the cytosol, about 80% of the amine groups are unprotonated, while at pH = 5 (endosome), the not protonated amine groups are 50% [72]. This prerogative allows PEI-DNA polyplexes to cause lysis of the endosome following swelling and to avoid lysosomal trafficking, which would lead to their degradation [72].

Remant et al. reported that the transfection capacity of PEIs increases as their molecular weight increases, but at the same time, there is an increased cytotoxicity caused by the aggregation of the polymeric complexes, which leads to trigger cell necrosis [72].

The optimal molecular weight of PEI is between 5 and 25 kDa. The type of structure, linear or branched, and the number and types of nitrogen atoms also determine the ability to condense DNA [72].

Typically, b-PEIs have been found to compact DNA better than l-PEIs of the same molecular weight. Furthermore, the complex stability is higher for complexes containing many primary amino groups, and many in vivo studies have demonstrated that b-PEI is more efficient as a vector for gene transfection than l-PEI.

Several chemical modifications to the polymeric structure have been attempted to increase the TE of PEI. The most important was to introduce PEG segments (PEGylation) into the PEI structure. This modification created a hydrophilic outer crown, reducing free PEIs’ tendency to self-aggregate and interact with plasma proteins and erythrocytes [73]. In different studies, polyplexes PEI/DNA were coated with PEGylated PEI and targeting ligands such as RGD and HIV-1 Tat peptides (H-Arg–Gly-Asp–OH), hyaluronic acid (HA), transferrin (Tf), epidermal growth factor (EGF), and folic acid (FA) [72,74,75,76]. Suk et al. grafted RGD or HIV-1 Tat peptides onto PEI/DNA complexes via coating technology using PEGylated PEIs [75]. The cellular uptake of the complexes in neuronal cells was significantly improved, as was the TE. The synergistic effect of targeting ligands and PEG spacers was investigated in in vivo gene delivery by Ogris et al. [76]. The authors reported efficient gene transfection in subcutaneous Neuro2A neuroblastoma tumor cells in syngeneic A/J mice upon intravenous injection of Tf-PEG-coated PEI/DNA complexes [76]. In the same study, EGF-PEG-coated PEI/DNA complexes were delivered via intravenous injection to target human hepatocellular carcinoma xenografts in SCID mice. In these studies, the gene expression levels in tumor tissue were significantly higher than those observed in other tissues. Additionally, the systemic administration of Tf-PEG-PEI/DNA complexes encoded TNF-induced efficient tumor necrosis and inhibited tumor growth. In another study, tetrary polymer DNA complexes (PoSC), made of DNA, PEI, polyspermine (PSP), and hyaluronic acid (HA), were reported [77]. Due to the synergistic effect of PSP and HA, PoSC complexes showed good compatibility with serum protein at various serum concentrations, high TE, and no toxicity at working transfection concentration [77]. 

There are various synthetic strategies for conjugating PEG and PEI covalently, including a two-step reaction. First, PEGs are activated using epoxy groups or isocyanates [78], and then the activated PEG is reacted with the amino groups of PEIs.

Figure 3 shows the activation of PEG with isocyanate groups and subsequent reaction with PEI to have PEGylated PEI.

While PEGylated PEI/pDNA complexes displayed prolonged blood circulation profiles compared with PEI/pDNA complexes, they demonstrated low interaction with cell membranes, internalization, and gene expression.

Xiong et al. managed to overcome this problem by eliminating the covalent bond between PEI and PEG. Particularly, aiming at bridging the gap between the positive attributes of PEG (prolonged particle circulation) and the positive attributes of cationic polymers (enhanced cell interactions), they performed noncovalent PEGylation of cationic particles via PEG-avidin/biotin-PEI achieving salt-stable complexes [79]. 

In addition to PEGylation, further modifications have been made in order to increase the transfection capacity of PEI, including quaternization of amines with methyl or ethyl iodides, alkylation of primary and secondary amines with 2-bromocholine, acylation of primary and secondary amines with amino acids, alkylation of primary amines with dodecyl or hexadecyl iodides, and quaternization of tertiary amines with hexadecyl iodide [27]. 

In particular, Thomas and Klibanov carried out such modifications on PEI2 and 25 KDa PEI to assess how they could affect the proton sponge capacity, the hydrophobic-hydrophilic balance, and lipophilicity [27]. Figure 4 reports the reactions carried out by the authors to introduce quaternary groups into the polymer scaffold of PEI using methyl and ethyl iodide (Figure 4). 

SAR studies using these modified PEIs, which were performed through transfection experiments on monkey kidney (COS-7) cells, reaffirmed the importance of the proton sponge effect [27].

The results obtained by the authors demonstrated that to have an efficient gene transfection, the presence of a high number of amino groups, which can be protonated in vivo without having a permanent positive charge, as in the case of quaternized derivatives, is necessary.

Modified PEI obtained by reacting PEI with 2-(bromoethyl)trimethylammonium bromide to create a structure bearing quaternary amino groups on the periphery but retaining secondary and tertiary amino groups capable of being protonated demonstrated enhanced TE compared with PEIs having only quaternized nitrogen atoms [27].

The unfavorable hydrophilicity-lipophilicity balance (HLB) of unmodified PEI was improved by introducing amino acids such as alanine and valine and alkyl chains by alkylation of the primary amino groups of PEI with dodecyl and hexadecyl halides.

While a proper HLB, which improved the TE, was obtained with alanine, the valine and alkyl chain derivatives showed too high lipophilicity.

In modified polymers, the efficiency of PEI25 in the presence of serum was doubled with a concurrent appreciable reduction in cytotoxicity, while the TE of PEI2 was enhanced 400-fold by increasing its lipophilicity.

#### 2.2.3. Linear Poly(amidoamines) (PAA)

Linear PAAs are polymers obtainable from the polymerization of cyclic linear aliphatic diamines with monomers of bis-acrylamide type [80] (Figure 5). 

Due to their polycationic and amphoteric nature, they have been studied for gene transfection, finding that their efficacy is comparable to that of lipid carriers such as Lipofectin or 70 KDa PEI [81]. More recently, PAAs have been synthesized starting from (4-aminobutyl)guanidine and 2,2-bis(acrylamido)acetic acid (Figure 6) [82].

Despite the high density of positive charge existing at physiological pH, the obtained PAAs have shown low hemolytic activity and low cytotoxicity while possessing a negligible elimination (clearance) by the endothelial reticulum without the need to be PEGylated. 

#### 2.2.4. Polymethacrylates (PMA) and Polyacrylamides (PAM)

The main exponent of this class of polymers used as vectors for gene therapy is poly [2-(dimethylamino)ethyl methacrylate] (PDMAEMA), which has a permanent positive charge at physiological pH. Although the synthesis was carried out by radical polymerization of 2-(dimethylamino)ethyl methacrylate triggered by ammonium peroxydisulfate (Figure 7), more recently, living PDMAEMA was prepared by ATRP starting from (2-dimethylamino)ethyl methacrylate. The reaction was carried out in bulk at 30 °C using CuCl/N,N,N,N,N-pentamethyldiethylenetriamine as a catalyst, and in the presence of the catalytic amount of tricaprylylmethylammonium chloride (AQCl) [83].

These polymers have been used in in vitro and in vivo gene delivery models. PDMAEMA has shown promising gene transfection activity due to its cationic character, and PEGylated PDMAEMA/DNA polyplexes demonstrated efficient brain-targeted gene delivery in mice [84].

The ability of PDMAEMA/DNA polyplexes to destabilize endosomes and dissociate from DNA in cytosol contributed to their TE [85]. 

To further increase the TE and the release of PDMAEMA-DNA complexes from the endosome by lysis, Funhoff et al. made structural modifications to PDMAEMA by introducing tertiary amine functions in the side chain to obtain the required buffering effect [86].

While these modifications led to a decreased toxicity, no improvement of cellular transfection was achieved, thus demonstrating that the “proton sponge” hypothesis is not valid for all polymers.

Also, it was reported that detrimental structural modifications can derive from the introduction of carboxylic residues in the side chain of PMAs, thus causing a decrease or loss of TE.

Furthermore, Figure 7 shows the structure of some acrylamide-type monomers containing the carbonate function in the side chain as a linker of tertiary amino residues of various kinds.

The polymers obtained from such monomers showed biodegradability, reduced cytotoxicity, and a transfection capacity higher than that of PEI 25 KDa [87].

##### 2.2.5. β-Cyclodextrins

In order to overcome the cytotoxicity of many non-viral polymeric vectors and to increase their biocompatibility, β-cyclodextrin residues were introduced into the structure of cationic polymers.

The synthesis of such polymers can proceed via polymerization of a bi-functional β-cyclodextrin monomer, such as (2-aminoethanthio)-β-cyclodextrin, with an additional bifunctional co-monomer, such as dimethylsuberimidate hydrochloride (Figure 8) [88].

The transfection obtained with polymeric vectors based on β-cyclodextrins was better than that obtained with PEI-based vectors.

It has also been shown that the length of the alkyl chain of the dimethyl-suberimidate spacing and the monomeric units of cyclodextrin greatly influenced the cytotoxicity of these vectors. In particular, cytotoxicity decreased with longer chains, probably due to a decrease in charge density [89].

The best compromise between low cytotoxicity and more efficient transfection was found for compounds with six to eight methylene group spacers.

Collectively, due to their good water solubility, biocompatibility, and extensive recognition ability of cyclodextrins (CDs), supramolecular delivery systems using CDs as building blocks have been widely studied by researchers [90,91]. Davis and co-workers reported on siRNA and the CD-adamantane complex for inducing an RNA interference (RNAi) mechanism, which can provide new and major ways of imparting therapy to patients. Such a complex was the first example of using supramolecular assemblies for siRNA delivery in clinical applications [92]. In another study, it was reported on a cationic CD-based supramolecular polymer that demonstrated capacity in DNA compaction and was able to interact with siRNA and allow its transfection [93]. CD systems for DNA delivery have made great progress, proving the feasibility of the supramolecular systems for gene delivery and their potential clinical application.

#### 2.2.6. Chitosans

Chitosans are polysaccharides obtained by deacetylation of chitin with the formation of polymer structures composed of units of D-glucosamine and N-acetyl-D-glucosamine linked by a β-(1,4)-glycosidic bond (Figure 8) [94].

Thanks to their biocompatibility, biodegradability, and overall charge at physiological pH, chitosans are considered non-viral vectors with great potential for the transport of genetic material [95].

Since the mid-1990s, Mumper et al. have studied chitosan polymers for gene transfection experiments [96].

Studies have shown that the molecular weight of polysaccharide derivatives can influence their transfection capacity, and chitosans with high molecular weights have been found to form more stable complexes with DNA than their lower molecular weight analogs [97].

Other important factors that profoundly influence the characteristics of chitosans and their TE are the degree of deacetylation, the pH, and the N/P ratio [97].

It has been observed that the TE in vitro and the stability of the chitosan/DNA complex can be improved by increasing the degree of deacetylation [97].

Curiously, in vivo studies have shown better activity for chitosans with medium rather than high degree of deacetylation. These conflicting data indicate that an effective transfection in vivo can be achieved, mainly by balancing the DNA protection from nucleases (more stable complexes) with the ease of release (fewer stable complexes).

A correlation between TE by chitosan complexes and pH has also been demonstrated. At pH > 7.5, the detachment of the nucleic acid from the polymeric complex occurs before its cellular uptake. At pH < 6.5, there is a significant cellular uptake of the complex but also a low capacity to release genetic material, probably due to a difficult release by the endosome. Therefore, the optimal pH is between 6.8 and 7.0 [97].

Figure 9 shows some chemical modifications of chitosans aimed at improving their TE.

In order to increase the cationic charge density, chitosan copolymers grafted with polylysine residues (chitosan-g-PLL) were prepared, which showed an excellent DNA complexing capacity, reduced toxicity and increased TE compared to PLLs or PEI alone [98] (Figure 9A).

Since simple alkylation with methyl iodide gave polymers that showed better TE but high cytotoxicity, chitosans were prepared with PEG segments, which were introduced by reacting PEGs activated with N-hydroxysuccinimide (NHS-PEG) with N-trimethylated chitosan [99] (Figure 9B).

Further structural modifications aimed at improving the buffer capacity of chitosan-based carriers have been shown in Figure 10.

In the first strategy (right side of the Scheme), chitosan was conjugated via amide bond with urocanic acid [(E)-3-(1H-imidazol-4-yl)prop-2-enoic acid] in different ratios, exploiting its NH_2_ groups in position 2. Thanks to the presence of the imidazolic ring, the resulting complexes showed a better buffering capacity, reduced toxicity, and excellent TE [100].

In the second strategy (left side of the Scheme), chitosan was used as an initiator molecule to polymerize aziridine, thus obtaining a chitosan-PEI copolymer endowed with TE like that observed previously [100], equal to that of the 25 KDa PEI, but better than those of the single PEI or the single chitosans [101].

#### 2.2.7. Dextrans

Dextrans represent an important class of saccharide polymers that are extensively applied in chemistry and biology. Structurally, they are glucans where the glucose units are linked with α-(1,6)-bonds and have short branches linked with α-(1,3)-bonds.

The most used dextran for gene transfection experiments is the diethylamine-ethyl-dextran (DEAE-dextran) [102]. More recently, the interest of researchers has focused on the dextran-spermine system, which was first synthesized via the oxidation of dextran with potassium periodate and subsequent reductive amination with spermine and sodium boron hydride (Figure 11) [103].

Dextran-spermine polymers with molecular weight of 6–8 KDa and 25–30% of spermine residues have shown a TE in vitro equal to Transfectam and DOTAP [103].

Also, in this case, the quaternization of the terminal nitrogen atoms has decreased the ability to release genetic material from the polymeric complex, probably due to interactions with the polymer being too strong [104].

### 2.3. Dendrimer Vectors

As already reported, dendrimers are highly branched synthetic polymers characterized by a monodisperse nature with a very regular spherical and globular structure with nanometric dimensions. Due to their structural characteristics, dendrimers have aroused strong interest in several sectors, and their potential in biomedical applications, including gene therapy, has been extensively investigated [1,7,13].

#### 2.3.1. Polyamidoamines (PAMAM)

PAMAMs are the most studied and used dendrimers as carriers of genetic material due to their experimented synthesis, their structural variety, and the commercial availability of some compounds of their family. They can be prepared by a divergent method performing reiterative Michael additions of a nucleophilic core such as ethylenediamine to methyl methacrylate, followed by amidation of the resulting ester with a suitable functional amine (Figure 12) [105]. 

This method’s main limitations include the possibility of having PAMAMs with structural defects caused by incomplete Michael addition reactions. Other undesirable side reactions comprehend retro-Michael reactions, intramolecular cyclization, and the solvolysis of terminal functional groups, also leading to non-ideal dendrimer growth and impurities, difficult to separate from intact dendrimers because of too similar physicochemical properties [106]. A convergent method was developed by Hawker et al. in 1990 to address these drawbacks, by which the dendrimer was synthesized from the outer surface to the inner core [107]. 

PAMAMs have been shown to have a high capacity to transfect various cell lines. In PAMAMs, the genetic cargo can be entrapped and protected within the internal cavities, bound to the peripheral functional groups, or even loaded by a combination of the two approaches. Cationic amino-terminated PAMAM dendrimers, because of their positive charge, have both the ability to bind the negatively charged nucleic acids and to promote efficient cellular uptake by endocytosis and/or via membrane pore formation [108]. PAMAMs have been extensively studied as vectors for nucleic acid-based therapies using pDNA and siRNA/antisense oligonucleotides forming complexes named dendriplexes [108]. 

Studies reported that the activity of PAMAMs in binding siRNA mainly depends on their generation [109], molecular flexibility, and pH. Although in an old study concerning transfection by PAMAMs, it was reported that the transfection activity is maximal using dendrimers of high generations (G5-G10) [110,111], it was demonstrated that while G4 PAMAMs possess good adaptability to siRNA, G6 molecules, behaving like a rigid sphere, demonstrated a consistent loss in the binding affinity. Expectedly, G5 PAMAMs showed a hybrid behavior, maintaining rigid and flexible aspects, with a strong dependence of their properties on the pH. In addition to their generation, the transfection activity of PAMAMs depends on the charge density of the dendrimer-DNA complex (dendriplex) and the dendrimer/DNA ratio (N/P), namely R.

The appearance/disappearance of some important properties was observed depending on the R-value.

For R values between 0 and 1, a good solubility of the dendriplex was observed, but only small changes in the original conformation of the DNA and ineffective compaction were found [112].

For values of R between 1 and 100, the formation of insoluble aggregates was observed due to the neutralization of the negative charges of the nucleic acids, while for R greater than 100, a resolubilization of the dendriplex occurred [112].

However, the main drawback of using PAMAMs remains their high cytotoxicity. Additionally, although PAMAMs are promising candidates for gene delivery, the lack of targeting and poor pharmacokinetics limit their clinical translation. Toxicity is mainly attributable to the PAMAM cationic surface charge, which increases with the PAMAM generation but is, paradoxically, necessary for a good TE. Moreover, the toxicity of PAMAMs is due to their low biodegradability [111]. To address these issues, chemical modifications have been attempted, including PAMAM end group modifications using amino acids [113,114,115,116,117,118,119], lipids [120,121,122], cyclodextrins [123,124], PEG [125,126,127], and others [128,129,130,131]. Fang et al. reported the partial modification of PAMAMs end groups with p-toluylsulfonyl arginine, which resulted in an increase of the TE while minimizing cytotoxicity [132]. The TE improvement could be explained by the additional possibilities of electrostatic, H-bonding, and hydrophobic interactions with both DNA and the cell membrane, thus determining high cellular uptake efficacy and endosomal escape capability. On the other hand, the lower charge density of these modified PAMAMs resulted in a lower cytotoxicity. PAMAM modifications by means of amino acids, including arginine and histidine, allowed the formation of stable complexes, thus improving cell uptake and buffering capabilities, respectively [118,119]. Smith et al. reported the synthesis of PAMAMs whose branches were hybridized with peptide segments (DendriPeps) [133]. They used G2 and G3 PAMAMs as host scaffolds and lysine or glutamic acid as “guest” amino acids, obtaining PAMAMs displaying additional primary amines or carboxyl functional groups [133]. Recently, peripheral modifications of PAMAMs to improve the delivery of messenger RNA (mRNA) were reported, including modification using lysine by amidation reactions in a regioselective, quantitative, and stepwise manner [134]. PAMAMs were also modified with p-toluylsulfonyl arginine, as previously reported by Fang [132], for its fusogenic properties and imidazole groups conferring extra buffering capacity to facilitate endosomal escape. The modified dendrimers were then complexed with Cy5-EGFP mRNA, providing small, reliable, and well-encapsulated positively charged dendriplexes capable of cellular delivery and translation of mRNA in several cell lines while maintaining a low cytotoxic profile [134]. 

#### 2.3.2. Poly(propyleneimines) (PPI)

As reported [2], the first synthesis of this class of dendrimers was executed using a divergent approach. In particular, it consisted of a repeated sequence of Michael additions between a primary amine and acrylonitrile followed by reduction with sodium boron hydride and cobalt (Figure 13).

The low yields recorded using Co(II) and NaBH_4_ as reductant reagents were improved using gaseous hydrogen and Co-Raney as catalysts under more controlled conditions. This modification allowed the large-scale synthesis of PPIs and their commercialization. Specifically, PPIs are comprised of poly-alkylamines with primary amine terminal groups and a scaffold including many tertiary propylene imines [135]. They are commercially available up to the fifth generation (G5) and are widely utilized both in biology and material science [136].

Particularly, the high number of basic amino functions in the periphery (pKa 9–11) are responsible for the interaction with DNA, while the more acidic internal tertiary amino groups (pKa 5–8) are not available to bind DNA but are able to act as “proton sponges” within the endosome [136].

Subsequent studies have shown that while all generations of PPIs are capable of condensing genetic material into compact particles, only those of higher generations provide water-soluble cationic dendriplexes, even if the PPI-DNA complexes of lower generations have shown less toxicity [137].

Important structural modifications have been made to improve the TE and reduce PPIs’ toxicity. Hashemi et al. reported the modification of a G5 PPI with C6, C10, and C16 alkanoate groups inserted as hydrophobic moieties [138]. The ethidium bromide exclusion assay proved the ability of modified carriers to condense DNA, while the transfection assay showed higher DNA delivery efficiency compared to that of pristine G5 PPI and Superfect^TM^ (Qiangen, Milan, Italy). 3-(4,5-Dimethylthiazol-2-yl)-2,5-di phenyltetrazolium bromide (MTT) and apoptosis experiments showed lower toxicity for modified carriers compared to unmodified PPIs [138]. Additionally, arginine-conjugated PPIs were prepared as non-toxic and efficient gene delivery carriers [139].

Lee et al. synthesized pseudorotaxane-terminated PPI dendrimers (Figure 9) by conjugation of PPI with compound A (Figure 14), in turn easily achievable from mono-carbobenzyloxy (Cbz)-protected diaminobutane. Upon deprotection and treatment with cucurbituril (CB), G1-G4 pseudorotaxxane-terminated PPI-based dendrimers were obtained [140].

Such PPI derivatives showed reduced cytotoxicity and gene transfection activity comparable to that of PEI, as well as the possibility of functionalizing the cucurbituril residues with amino groups. 

##### Dendritic Polylysine (DPL)

DPLs were born to overcome the drawbacks demonstrated by linear PLLs, including the high toxicity. Denkewalter et al., at the beginning of the 1980s, employing solid-phase synthesis to generate a G10 polypeptide dendron, performed the first synthesis of DPLs by a divergent growth approach [141] 

Particularly, the authors linked a lysine monomer Boc-protected and activated by a p-nitrophenyl ester to a bidirectional asymmetric nucleus (obtained by condensation of L-lysine and benzydrylamine). Subsequently, upon acidic deprotection, a first-generation DPL capable of undergoing additional coupling steps to give higher-generation structures was achieved (Figure 15). 

These structures were revisited by Posnett et al., who reported the divergent construction of a third-generation unsymmetrical polylysine dendron using conventional solid-phase peptide synthesis (SPPS) [142]. DPLs have also been achieved on acrylamide-PEG Co-polymer (PEGA) and Tentagel resins [143,144,145,146]. The convergent solid-phase synthesis of DPLs on silica support has also been reported [147]. Moreover, the divergent growth of G4 polylysine dendrons was accomplished using PEG as a hydrophilic tail, facilitating the product’s isolation and purification [148]. The convergent growth approach has also been employed to construct G4 polylysine dendrons decorated with eight mannose or galactose groups and a single fluorescein isothiocyanate (FITC) dye [149], while supramolecular structures based on polylysine dendrons were reported by Hirst et al. [150]. Supramolecular chemistry was also successfully employed to obtain polylysine dendrimers from dendron derivatives. The dendrimers were assembled using the covalently attached crown ether and diammonium compounds and disassembled using potassium cations.

Studies of gene delivery in this class of dendrimers have revealed interesting properties. Okuda et al. investigated the relationship between the structure of the DPL-DNA dendriplex and its TE. Dendriplexes with DPL of higher generations showed greater transfection capacity. Particularly, G6 DPLs exhibited the highest activity, thanks to the stability of the complexes they form, which improved the endosomal escape and the availability of RNA polymerase within the nucleus despite decreased cellular uptake [151,152].

Several modifications to the DPL structure have been thoroughly investigated to further improve TE, including the replacement of the terminal lysine residues of DPL with arginine [153] or histidine [154]. Arginine derivatives showed significant DNA binding and transfection capacity, up to 3–12 times higher than unmodified DPL. On the contrary, histidine derivatives showed little complexation with genetic material and no gene transfer capacity. 

## 3. Dendritic Architectures Derived from 2,2-Bis(hydroxymethyl)propanoic Acid (b-HMPA)

It has been reported that, historically, in 1952, it was Flory who foresaw the possibility of obtaining highly branched polymeric structures from monomers of the AB_n_ type, characterized by a high number of functional groups at the periphery [155].

Years later (1978), polymeric structures of this type were defined as “cascade molecules” [156].

Subsequently, other examples of this new class of polymers appeared in the chemical literature associated with the terms “arboreal” and “dendrimer” [157]. Today, only the latter term is universally accepted.

The molecular structure of a dendrimer can be schematically described as follows. Starting from a multifunctional core (usually di-, tri-, or three-functional), successive layers (generations) are obtained from the reactions of branched monomers of the type AB_n_ with n ≥ 2 branch out.

Figure 10, our image in which we have reproduced an example reported by Walter et al. [158], shows the structure of a monomer of the AB_2_ type, as well as of the G4 dendron and G4 dendrimer which can be derived from it.

The class of dendrimers is vast and includes various types of molecular architectures, all with great prospects for use in various fields, from optics to biology. The preparation of dendrimers requires the application of “robust” chemical reactions and careful procedures of isolation and purification, as well as sophisticated techniques of characterization involving expensive equipment. A family of dendrimers that has attracted particular interest among researchers is derived from the AB_2_ monomer 2,2-bis(hydroxymethyl)propanoic acid (b-HMPA, compound 1, Figure 11). The molecule is a trifunctional aliphatic, pro-chiral compound that has a molecular weight of 134.06 g mol/L. b-HMPA contains two primary alcohol groups (B_2_) and a carboxyl group (A), is easily commercially available, even in large quantities at low cost, and can be used for the preparation of a wide variety of dendritic molecular architectures [159].

Furthermore, the polyester systems derived from b-HMPA have shown good biocompatibility, making them particularly attractive materials for applications in the biomedical field [14]. Figure 12 schematically shows the various types of dendritic structures that can derive from bis-HMPA [160].

Monodisperse systems include homo- and hetero-functional dendrons and dendrimers, while hyper-branched polymers and hybrid dendritic linear polymers belong to polydisperse systems. Furthermore, hybridizations of organic and inorganic surfaces were carried out using b-HMPA, obtaining highly functionalized dendronized surfaces of great potential in the chemistry of materials.

### 3.1. Strategies Used for the Synthesis of b-HMPA-Derived Dendrimer Systems

Different strategies have been developed for the synthesis of dendrimers, which can be divided into conventional approaches (divergent and convergent growth (DG and CG) approaches) and revisited or unconventional approaches, including accelerated synthesis. The “click chemistry-based approaches” also represent an advantageous type of accelerate synthesis (Table 1). 

#### 3.1.1. Conventional Approaches

In these approaches, the synthesis is based on iterative steps of protection and activation of selected functional groups followed by growth. It is important that the reactions can proceed with high yield while the generations grow.

##### Divergent Growth

This approach is also called inside-out since the growth of the dendrimer starts from a multifunctional inner core C_n_ (n ≥ 2) and develops outwards. The method involves the use of AB_n_ (n ≥ 2) monomers, where A is an activated group, while B is inactivated/protected functions to allow controlled growth. The active functions of core C are selected to be able to react with the activate group (A) of the monomer. Depending on the number of functional groups of the core, a correct excess of the AB_n_ monomer must be used to achieve the first generation. To continue the growth, the B functions of the obtained first-generation (G1) dendrimer must be activated/deprotected to allow a new reaction with the groups A of a furthers excess of the monomer AB_n_ to afford the following generation of dendrimer and so on.

The repetition of the abovementioned steps leads to a progressive increase in the generations of the dendrimer and an increase in the number of external functional groups. Once the desired generation has been achieved, the peripheral functional groups are available for further post-functionalization reactions with a wide variety of reagents.

A drawback of this strategy is the risk of building defective dendrimers, which are difficult to separate from the perfect ones due to the small structural differences and physicochemical properties of the precursors and the products. The structural defects may originate from the incomplete conversion of all the peripheral groups, mainly due to the steric hindrance that increases with the growth of the generations.

By this synthetic strategy, Ihre et al. reacted to the b-HMPA derivative benzylidene anhydride 2 (Figure 11) with a triphenol core. Subsequently, reactions of esterification to increase the generations, followed by deprotection reactions of the external benzylidene groups by hydrogenolysis, led to the obtainment of a six-generation (G6) dendrimer with 192 free peripheral hydroxyl groups in 95% yield. As an example, Figure 16 shows the procedure up to the G5 dendrimer [164].

Following a similar strategy but using the anhydride shown in Figure 11, a G4-dendrimer was built divergently, starting from a G2-dendrimer having the same triphenyl core in eight consecutive passages [165].

##### Convergent Growth

Historically, in the 1990s, Hawker and Fréchet introduced a convergent growth approach as an alternative route for dendrimer synthesis [107]. The method was based on the realization of perfectly branched dendrons (dendrimer segments) peripherally protected, which were finally coupled with a central core upon the activation of their focal point (see Figure 10). Following the activation of peripheral protected functional groups, the post-functionalization of the obtained dendrimer with several molecules was possible. The first application of a convergent strategy using b-HMPA led to the obtainment of a G4 dendrimer with a triphenol core (Figure 17) [166].

In the convergent technique, monitoring the growth of the various generations of dendrons is easier to carry out, and the purification procedures are less laborious due to the greater structural diversification of dendrons and dendrimers compared to the divergent method. Regardless of the strategy applied, NMR spectroscopy, in particular the ^1^H NMR technique, was very effective in controlling the growth of dendrons and dendrimers. At the same time, mass spectrometry, in its MALDI-TOF version, constituted the most important investigative tool for the confirmation of the final dendrimer structure.

#### 3.1.2. Revisited Methods: Accelerated Approaches

As reported above, the synthesis of dendrimers by conventional approaches is long and laborious, and the probability of introducing defects within the dendritic architectures is high. Starting from an AB_2_-type monomer, the synthesis of a G4 dendrimer requires at least eight steps. Moreover, to obtain dendrimers of high generation, increasing excesses of reagents are necessary to ensure the complete replacement of all reactive groups. Accelerated strategies have been developed to address these issues, aiming at reducing the number of reaction steps and the consumption of material. After synthetic efforts, original strategies have been developed, with crucial points reported in Table 2.

##### Hyper Monomer

In the hyper-monomer strategy, monomers with more than two functional groups, e.g., AB4 monomers, were used instead of the traditional AB_2_ ones, where the dendrimer structure can be built, as reported in Figure 13. 

While using an AB_2_ monomer, eight passages are required to obtain a G4 dendrimer; only four passages are required when an AB_4_ monomer is used [161]. Wooley et al. reported the reaction of a third-generation dendron (D3) corresponding to an AB_8_ hyper-monomer, with a second-generation dendron (D2), corresponding to an AB_4_ hyper-monomer obtaining a fifth-generation dendron D5, albeit with poor yield due the steric hindrance and the low efficiency of the reaction used [167]. Collectively, this strategy’s main criticism consists of synthesizing the hypermonomer, which, in any case, requires a multistage procedure.

##### Two-Stage Convergent Growth (TSCG) 

This approach achieved generational growth by functionalizing a low-generation dendrimer (hyper core) with low-generation dendrons. Xu et al. succeeded in synthesizing G4 phenylacetylene dendrimers by coupling a D2 dendron with a G2 dendrimer [168]. Although the final dendrimers were obtained in one step, the preparation of the dendrons and the hypercore required four synthetic steps each. Consequently, the fabrication of the G4 dendrimer was realized after a total of nine steps. In this regard, this method should be limited to the preparation of higher-generation dendrimers that cannot be obtained with conventional methods.

##### Double Exponential Growth

In this strategy, protected or deactivated low-generation dendrons were prepared, which reacted with each other upon activation at their focal point or periphery. By this approach, Ihre et al. synthesized a b-HMPA-based G4 dendrimer containing 48 hydroxyl groups at the periphery (Figure 18) [169].

Using a conventional growth approach, a D2 dendron bearing two acetonide groups at the periphery and a benzyl group at the focal point was first synthesized. Then, upon selective deprotections at the focal point and periphery, two dendrons were obtained, which reacted with each other, thus providing a fully protected D4 dendron.The activation of the D4 focal point by removal of the benzyl group through hydrogenolysis, followed by reaction with a triphenol core and the removal of the acetonide protections, led to the b-HMPA-based G4 dendrimer. The main drawback of this approach is the drastic decrease in yield with the increase of dendrimer generations, which makes it unusable for synthesizing high-generation dendrimers. Over the years, the polyester-based dendrons deriving from b-HMPA have been attached to various types of cores and have been functionalized to the surface with molecules selected on the basis of the desired application. Malmstrom and Hult [170] reported the synthesis and characterization of dendrons linked by a spacer to a porphyrin kernel, chosen since useful in various fields, such as optoelectronics, sensors, and biomedicine for vehicle of biological molecules (Figure 14).

Fréchet [171] synthesized b-HMPA-based dendrimers with pentaerythritol core functionalized on the surface with cyclic carbonates, which, in the presence of amines, such as 2,2-di-methoxyethylamine opened, releasing the hydroxyl groups which were further functionalized with propargyl bromide (Figure 15).

Moreover, G1 dendrons attached to the pentaerythritol core and functionalized on the surface with PEG-amino acids chains were used as carriers for anti-tumor drugs [172].

##### Procedures via Click Chemistry

An important breakthrough in the synthesis of dendrimers was due to the emergence of Click Chemistry. This term, coined by Sharpless in 2001 [162], regards the coupling reactions characterized by a high “thermodynamic thrust”, such as the copper-catalyzed azide-alkyne cycloaddition (CuAAC), the Diels-Alder (DA) reaction, or the thiolene coupling (TEC) reactions (Figure 19).

Click reactions are usually performed at room temperature and require simple purification steps such as extraction or filtration. They are highly selective, with yields close to 100%, work well in the presence of a wide variety of solvents, including water, and are compatible with a large number of functional groups [163]. Hawker, Fréchet, and Sharpless showed that the Cu(I)-catalyzed 1,3-dipolar azide-alkyne cycloaddition (CuAAC) yielding a triazole product, could be a very appealing method to synthesize complex polymeric structures such as dendrimers [173]. Consequently, the application of CuAAC has become an important tool for preparing functionalized dendrimer macromolecular architectures. Due to the efficiency and selectivity of the CuAAC reaction, Antoni et al. synthesized a G4 dendrimer starting from monomeric units of the type AB_2_ and CD_2_ and a triphenol core [174]. Specifically, Figure 16 shows the structures of the monomers used.

The halogen atoms in the AB_2_ monomers were initially used in etherification or esterification reactions with the triphenol core, while the C-C triple bonds of CD_2_ monomers were reacted with the azide functions to generate the triazole systems. The resulting hydroxyl functions at the periphery were reacted with the AB_2_monomers and so on. In this way, it was possible to synthesize G4 dendrimers in just four steps and with excellent overall yields [175]. Figure 17 summarizes schematically the designed synthetic sequence.

### 3.2. Hetero Functional Dendrimer Systems

Initially, dendrimers containing only one type of function at the periphery were synthesized, and their structural variety was vast. With the growth of applications of dendrimer systems, the researcher considered the presence of only one type of peripheral function as a limitation to their possible applications. Consequently, ways have been explored to synthesize more sophisticated dendrimers that could express more than one type of function. Different synthetic approaches have been devised and implemented to obtain heterofunctional dendrimers (HFD) with different functions arranged in different positions within the dendritic architecture, both internally and externally [176]. The first example of HFD synthesis starting from b-HMPA was reported by Gillies and Fréchet [177]. Particularly, by combining the use of benzylidene anhydride (2) and b-HMPA acetonide protected, the authors succeeded in preparing a G3 hetero-functional dendrimer in 12 consecutive steps, thus evidencing the major synthetic complications existing for obtaining HFDs, which greatly limited their accessibility. A simplified strategy to obtain HFDs from b-HMPA was proposed later by Goodwin et al. [171]. As previously reported, reacting a b-HMPA-based G1 dendrimer bearing eight free hydroxyls with b-HMPA protected as carbonate and by applying a divergent growth approach, a G2 dendrimer containing eight cyclic carbonate groups in the periphery was achieved. Then, treating it with a series of amines, thanks to careful and programmed use of precise stoichiometric ratios, an HFD was synthesized bearing eight hydroxyls for further functionalization and eight acetals, thus alternating functions in just four steps [171]. Recently, García-Gallego et al. synthesized a new class of internally and externally functionalized multipurpose dendrimers based on b-HMPA by the elegant and simple design of AB_2_C monomers, obtained from two traditional AB_2_ monomers (Figure 20) [176]. 

Particularly, utilizing fluoride-promoted esterification (FPE), straightforward layer-by-layer divergent growth up to the G4 dendrimers displaying 93 reactive groups divided by 45 internal and 48 external functionalities, was successful in less than one day of reaction time. Post-functionalization through Click reactions and cross-linking into multifunctional hydrogels were demonstrated.

#### Hybrid Dendritic Systems

In the continuous search for polymeric structures with new properties, chemists have succeeded in preparing hybrid systems that derive from the presence of linear polymers and dendrimers in the same molecular structure. Their preparation required the intervention of chemists experts in both organic chemistry and macromolecular chemistry due to the complexity of the synthetic processes, the products’ purification, and their characterization. Linear dendrimer-polymer hybrid systems were first investigated by Fréchet et al. in the early 1990s, and the interest in these hybrid systems continues to be high due to the unique properties that some of them have shown both in solution and in the solid state [178,179,180,181]. Figure 18 pictorially shows some hybrid structures where the linear polymeric and dendrimer parts are visually recognizable.

Also, in this case, b-HMPA was used as an AB2-type monomer unit for constructing hybrid dendritic structures. The most widespread strategies for the synthesis of this new class of polymers based on b-HMPA have been described by different authors using graft-from, graft-to, and macromonomer approaches [182]. Table 3 summarizes these techniques.

All the most commonly used polymerization techniques have been successfully employed in this field. Reactions such as cationic and anionic polymerizations, ring-opening metathesis polymerizations (ROMP), radical polymerizations, atom transfer (ATRP), and chain transfer by reversible addition (RAFT) polymerizations, which guarantee tight control over the degree of polymerization and the distribution of molecular weights, were preferred [182]. A particular hybrid system different from those above described is represented by the “hybrid star-like dendritic systems” (star-like) in which a dendrimer plays the role of the macro-initiator of the polymerization (core-first approach) or the support for the attachment of preformed polymers (arm-first approach) (Table 3). In both approaches, the polymeric chains branch off from the dendrimer to give a structure similar to a star, where the number of arms depends on the number of peripheral functional groups of the initial dendrimer structure. A core-first approach has been reported by Hedrick et al., who used the hydroxyls of G1 and G2 dendrimers derived from b-HMPA to promote the ring-opening polymerizations (ROP) of the ε-caprolactone (ε-CL) and lactic acid (L-LA) [184]. In an extension of this work, both the monomers (ε-CL and L-LA) were polymerized at 110 °C in the presence of G1-4 dendrimers using Sn (Oct)_2_ as catalyst [185]. Concerning the arm-first approach, Nyström et al. performed a direct coupling via esterification between the carboxyl function of PEG polymers having a molecular weight of 5 or 10 KDa and commercial dendrimers of b-HMPA, namely Boltorn H30 and H40 [186]. In both approaches, all end groups at the periphery of the core should initiate the polymerizations or should be coupled with the linear polymers. On the contrary, it would be very difficult to separate the non-functionalized molecular systems from the functionalized ones and to characterize them through analytical techniques such as NMR and MALDI-TOF. Additionally, the preparation of dendritic-linear block co-polymers constituted by linear polymeric chains functionalized at the ends with dendritic segments has been reported. In one of the first examples of these structures, benzylidene anhydride (2) was initially coupled via ester linkage to mono-, di-, and tetrahydroxylated PEG polymers. Subsequently, a divergent growth of dendrons was performed starting from the functionalized PEGs [164]. In another study, the synthesis of a series of new photo-addressable linear-dendritic di-block copolymers was achieved by a mixture of double-stage convergent method (DSCM) and click chemistry. The prepared materials were composed of poly (ethylene glycol) (PEG) and of G4 b-HMPA-based dendritic aliphatic polyesters functionalized at the periphery with mesogenic and photochromic cyanazobenzene units [187]. In another example, the use of the CuAAC reaction allowed the connection between dendrons derived from anhydride (3) functionalized with terminal alkyne residues and PEG polymers, first properly activated with azide groups (Figure 21) [187].

### 3.3. Dendronized Surfaces

A sector of dendrimer chemistry with considerable potential in nanostructured materials is dealing with the immobilization of dendrimer structures on solid surfaces to provide the so-called “dendronized surfaces”. Dendritic structures can be immobilized by simple absorption or by creating a covalent bond between the dendrimer and the surface. In turn, the dendritic structure could be either normally self-assembled on a surface, forming monolayers or multilayers, or preformed dendrons could be chemically linked to the surface through their focal point or by their peripheral groups. The graft-to and graft-from strategies were also applied for the functionalization of the surfaces. Montanez et al. carried out the dendronization of cellulose surfaces activated with the azide function by reacting them with b-HMPA-based dendrons functionalized with acetylene residues up to the fifth generation (5G) via CuACC click reaction [189]. The terminal hydroxyl groups of the dendrons were further functionalized by a graft-from approach with an AB_2_C monomer (Figure 22). The resulting surface was finally functionalized with carbohydrate residues, antibiotics, dyes, and glycol chains.

Other examples of b-HMPA dendronized surfaces included silanized silicon oxide [190], clay [191], and calcium carbonate [192] surfaces. Moreover, the dendronization of gold surfaces with b-HMPA dendrons has been reported [193,194]. In particular, PEG chains functionalized with SH groups at one end and OH groups at the other, which were first immobilized on the gold surface by means of the SH groups. Then, the terminal OH groups of the PEG chains were reacted with the anhydride (3) to divergently grow b-HMPA-based dendrons up to the fourth generation (G4). Figure 23 shows the strategy adopted.

### 3.4. Applications of Dendritic Materials Derived from b-HMPA

Dendritic materials derived from b-HMPA, thanks to the reliability of the synthetic procedures and the possibility of post-functionalization and hybridization, represent a unique class of polymers destined to find increasing uses in various fields, including the biological areas due to their biocompatibility and biodegradability. A large variety of carefully selected bioactive molecules have been successfully covalently linked to such dendritic structures, including carbohydrates (mannose, sialic acid, β-D-lactopyranose), amino acids (cysteine), peptides (glutathione or GSH), anticancer agents (doxorubicin) [195,196,197,198] and antibiotics (amoxicillin) (Figure 19).

More recently, b-HMPA-based dendrimers were functionalized with several amino acids, single or in a mixture of up to four different types, with dipeptides [1,7,13,26] and natural antioxidant molecules such as gallic acid [14]. To carry out the post-functionalization reactions, peripheral hydroxyl groups or carboxylic acid focal points were used as such or were converted to reactive intermediates such as alkenes [199,200], acetylenes [201,202] and p-nitrophenyl carbonates [177], allowing for the further introduction of polyethylene glycol chains (PEG) in order to increase hydrophilicity. Additionally, amine functional b-HMPA dendrimers decorated with β-alanine were evaluated for their antimicrobial activity and degradability. The second generation β-alanine-functionalized b-HMPA dendrimer, with 12 positive charges, inhibited the growth of bacteria *Escherichia coli* while being nontoxic to cells at the same concentration [203].

## 4. Our Experience

As illustrated in the previous Sections, b-HMPA has been used as a very versatile monomer for the construction of dendrimer systems with various architectures. The aforementioned structural variety, combined with their biocompatibility due to the ester scaffolds, places these systems in the first positions among the materials suitable for applications in the biomedical field.

On these considerations, in our laboratory, research activity has been developed aimed at obtaining dendrimers derived from b-HMPA functionalized with residues of amino acids, natural and not, to explore their effectiveness as vectors of genetic material with a view to gene therapy applications. This articulated research can be divided into four stages, and the prepared dendrimers have been classified as dendrimers of the first, second, third, and fourth age groups (we have not used the term generation to not create confusion with the generation of dendrimers themselves). Table 4 provides a schematic description of cationic polyester dendrimers based on b-HMPA developed in our laboratory and their main characteristics.

### 4.1. First Age Group Dendrimers

In this first step of our research concerning the synthesis of cationic dendrimers as possible carriers for gene delivery, starting from 2,2-bis(hydroxymethyl)propan-3-ol (t-HMPO) as core and b-HMPA as AB_2_ monomer, we have first prepared a D4 dendron (D4(A)) by a double-exponential grow approach (Figure 24). Subsequently, a G4 dendrimer equipped with 48 peripheral hydroxyl groups (G4OH) was prepared using a convergent approach (Figure 25). 

G4OH was then used for grafting different tert-butyloxycarbonyl (Boc)-protected amino acids (Gly, MGly, GABA, MGABA, Lys, or His), and after removal of Boc groups in acidic conditions, six G4 dendrimers with 48–96 cationic groups were achieved. Then, we used G4OH as a “hyper core” on which 1st and 2nd generation dendrons (D1 and D2) pre-functionalized with Boc-amino acids (Gly, MGly, GABA, Lys, His, Lys+His) were attached, affording after deprotection, nine new dendrimers with 96–384 cationic groups. For example, Figure 26 reproduces the synthetic approach followed to prepare the Boc-protected D1 dendrimers functionalized with lysine (Lys) and histidine (His), while Figure 27 used to perform both the direct functionalization of G4OH with lysine to achieve the G4 lysine dendrimer and the functionalization of G4 with D1-lysine to achieve the G5 lysine dendrimer. 

Collectively, a library of 15 polycationic homo- and hetero-dendrimers in the form of hydrochloride was obtained [26]. Their structures and composition were confirmed by NMR analysis and by experimental molecular weight computed by volumetric titration. Their buffer capacity, determined by potentiometric titrations, and results obtained from cytotoxicity assays and tests of binding with both pDNA and siRNA were very satisfactory [26].

### 4.2. Second Age Group Dendrimers

At this stage, starting from the same core previously used and b-HMPA, we prepared G4OH, which was functionalized with arginine alone, with a mixture of arginine and lysine, with a mixture of arginine and O-methyltyrosine and with the dipeptide arginine-glycine [13]. Secondly, using G4OH as a hyper core having 48 OH groups, we esterified it with b-HMPA acetonide protected, thus achieving, after acidic deprotection, a G5 dendrimer having 96 peripheral hydroxyls (G5OH). G5OH was functionalized with Arg alone, the dipeptide ArgGly, and the mixture Arg+Lys [13]. A total of seven polycationic dendrimers were finally obtained as hydrochlorides. As in the previous cases, their structures and composition were confirmed by NMR analysis and by experimental molecular weight computed by volumetric titration [13]. Their buffer capacity determined by potentiometric titrations was higher than that for some G4 PAMAM derivatives taken as reference. Different from the first-age group dendrimers, all these contained arginine. The introduction of arginine residues appeared particularly attractive because it is known that arginine-rich peptides are widely used as internalizing peptides for the delivery of nucleic acids [204,205] and siRNA [206]. Furthermore, examples of dendrimers already exist (although of a different nature than those derived from b-HMPA), containing arginine, which is demonstrated to protect siRNA by RNAase, to improve its cellular uptake, and high TE [207]. These characteristic properties have been attributed to the ability of protonated arginine residues to promote interaction with negatively charged residues, such as phosphate, sulfate, and carboxylate, present on cell membranes [204,205]. Additionally, methoxy tyrosine residues were introduced to confer dendrimers a certain hydrophobicity, which could promote the interaction with the cell membrane. As reported, the integration of tyrosine trimers in oligomers affected the hydrophobicity and stabilized the polyplexes formed with nucleic acid through the π–π interaction between the aromatic ring of tyrosine moieties. Oligomers terminally functionalized with tyrosine trimers and cysteines demonstrated favorable polyplex stability and enhanced gene transfer ability in most cases [208] 

### 4.3. Third Age Group Dendrimers

The third class of dendrimers synthesized in our laboratory comprehended only two G4 dendrimers but with a very complex cationic shell made of four different amino acids, including arginine, lysine, methyl-glycine (MG), and di-methyl-glycine (DMG) [7]. To this end, we prepared G4OH and developed two successful multi-step procedures to decorate it with the selected amino acids in a ratio as close as possible to the desired one, leaving the lowest possible number of hydroxyls unfunctionalized. The type of amino acids was chosen to have all possible varieties of amine groups (primary, secondary and tertiary) in the periphery for an improved buffer capacity and the guanidine one for the previously reported reasons. The inertia of DMG in the reaction of esterification with G4OH was solved using the α-bromo acetic acid as an esterifying agent, followed by a reaction with dimethylamine to remove brome and introduce the dimethylamine group [7]. After removing Boc protecting groups, a very satisfactory buffer capacity was achieved for the hydrochloride dendrimers. Their structures and peripheral composition were confirmed by NMR analysis and experimental molecular weight computed by volumetric titration. Cytotoxicity experiments performed on two model cell lines (B14 and BRL cells) demonstrated that the prepared dendrimers did not significantly affect cell viability, thus establishing a level of cytotoxicity significantly lower than that of PEI 25K when administered at the same concentration [1,209].

### 4.4. Fourth Age Group Dendrimers

The goal of the fourth stage of our research on b-HMPA-based cationic dendrimers was the preparation of a new class of amphiphilic hydrolyzable dendrimers potentially suitable for biomedical applications, including gene delivery [1]. In our project, the future dendrimers should have contained the least possible number of non-functionalized peripheral hydroxyl groups, a hydrophobic tail for a better HLB, they should have been equipped with a varied range of protonated amino groups for an enhanced buffer capacity, and they should have been endowed with low levels of cytotoxicity. To this end, a fully protected D2 polyester dendron was prepared using b-HMPA as an AB_2_-type monomer. In contrast, two hypercores were prepared to graft acetonide-protected b-HMPA on t-HMPO and pentaerythritol cores, previously equipped with a C-18 hydrocarbon chain as hydrophobic moiety [1]. Then, a G2 amphiphilic dendrimer was achieved by esterifying t-HMPO with the D2 dendron acetonide protected and activated at its focal point by a convergent approach. Finally, by a two-stage convergent growth (TSCG) strategy, two G3 dendrimers were obtained by reacting the two hypercores with the D2 dendron, which was opportunely activated/protected. After the removal of the acetonide-protecting groups, the obtained polyhydroxylated amphiphilic scaffolds were successively functionalized with one or more amino acids, as described in Table 4 [1]. In such a way, polycationic dendrimers were achieved in which, as a novelty, the usual inner polyester structure known for its biocompatibility was equipped with a fatty acid hydrocarbon chain as a hydrophobic residue for an improved HLB [1]. As reported in the literature, oligomers modified with hydrophobic blocks or bearing covalently linked amphiphilic chains demonstrated higher stability, higher capacity to enter cells, and endosome escape. A set of peripheral amino acids, which included L-arginine (Arg), L-lysine (Lys), N,N′-dimethylglycine (DMG), and N-methylglycine (MG), also mixed together, provided basic centers protonatable at different pKa values and optimal buffer capacity. In cytotoxicity experiments performed on two model cell lines (B14 and BRL cells), all the amphiphilic compounds only marginally affected cell growth since viability values were in the range of 79.9–111.7% for B14 cells and 79.6–105.8% for BRL cells [1]. They were less cytotoxic than dendrimers previously reported not equipped with the C-18 chain. Furthermore, the achieved amphiphilic dendrimers were found to be much less cytotoxic than PEI25K when administered at the same concentration, which determined the survival of only 30–40% of two cell lines [209]. Such dendrimers include all the features that make them look like new promising synthetic materials with a high potential for nanomedicine applications.

## 5. Commercially Available b-HMPA-Based Dendrimers

Table 5 shows the dendrimers derived by b-HMPA that are currently commercially available. Such dendritic polymers are biodegradable, have low cytotoxicity, and can be used in theranostics, biosensors, optics, adhesives, and coatings [210].

## 6. Conclusions

Modern medicine is focused on finding new therapeutic options to cure health diseases against which standard treatment methods fail or provide unsatisfactory results. Although gene therapy is still a growing medical field and such technology is still in its infancy, it represents one of the strategies that could potentially meet this goal. The main challenge for scientists in this sector consists of searching for safe and effective vectors of genetic material that are possibly non-toxic to the cells of treated individuals and capable of efficient transfection. Viruses-based transfection reagents, such as Lentiviral vectors, have been successfully used in clinical settings to treat hemoglobinopathies, and commercial Libmeldy was employed for the ex vivo gene therapy of children with metachromatic leukodystrophy. Remarkable results in clinical trials have been reported in the treatment of spinal muscular atrophy with adeno-associated viral vector-based gene therapy using Zolgensma. Approved gene therapies embrace Luxturna to treat RPE65-mediated inherited retinal dystrophy, Glybera to cure familial lipoprotein lipase (LPL) deficiency (LPLD), while Roctavian (BMN 270) promising as therapeutic for hemophilia A, is awaiting approval from the Food and Drug Administration. However, adenoviruses can cause severe innate and humoral immune responses, which can be dangerous, mainly in immunocompromised patients. Moreover, most people already have antibodies to adenoviruses, which can nullify the effects of gene delivery. In this regard, the use of synthetic carriers has been intensively developed in recent years, with evident success. In this paper, a description of gene therapy and the main cationic materials proposed for transporting and delivering genetic materials has been reported to give readers the necessary information. Then, due to their degradability in vivo, we have focused on dendrimers derived by b-HMPA. The history of the birth and development of the main b-HMPA-derived dendrimer scaffolds has been reported. The foremost synthetic strategies and the possible dendron and dendrimer architectures that can be prepared using b-HMPA have been reviewed. Also, a Table has been provided to collect the commercially available b-HMPA-based dendrons and dendrimers usable for further functionalization to achieve cationic materials for gene delivery application. Finally, thanks to a personal experience in the synthesis of b-HMPA-based dendrimers, our contribution to this field has been described. In particular, we have enriched this work by reporting about the b-HMPA-based derivatives peripherally functionalized with amino acids we have prepared in recent years. The amino acids-modified b-HMPA-based dendrimers developed by us have demonstrated the ability to bind pDNA and siRNA and low levels of cytotoxicity, thus representing interesting materials for further studies on their application in gene delivery and nanomedicine.

## Data Availability

All data concerning this work are included in the main text and in the references.

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
