# Peer review of "Cationic Materials for Gene Therapy: A Look Back to the Birth and Development of 2,2-Bis-(hydroxymethyl)Propanoic Acid-Based Dendrimer Scaffolds"

_ijms, 2023, doi:10.3390/ijms242116006_

Round 1

Reviewer 1 Report

Comments and Suggestions for Authors

Abstract: The abstract should be revised to include more specific information. It is recommended to modify the abstract section to provide a clearer overview of the manuscript.

What are the unique characteristics of b-HMPA-derived dendrimers and their potential applications in gene therapy and biomedical fields?

In the Introduction section, the authors are encouraged to state informative objectives to guide the readers.

Figures should be thoroughly polished to improve the overall presentation of the manuscript. Ensure that text in figures is clear and consider making color changes where appropriate.

How have b-HMPA-based dendrimers evolved over time, and what is the specific focus of the paper regarding amino acid functionalization?

Please include the reaction mechanism in a schematic form to enhance the visual representation of the process.

What is the primary objective of gene therapy and why are nonviral cationic materials, such as dendrimers, considered as alternatives to viral vectors in this context?

The conclusion should be rewritten to be more concise and provide an overarching summary of the findings and their significance. for example what is the main contribution of the authors to this field, and how does it relate to the broader context of gene therapy and dendrimer-based materials?

Replace older references with more recent ones to ensure the relevance and currency of the sources.

Author Response

Abstract: The abstract should be revised to include more specific information. It is recommended to modify the abstract section to provide a clearer overview of the manuscript.

We thank a lot the Reviewer for his/her suggestion. As asked the abstract has been fully revised and enriched with more specific information. A clearer overview of the manuscript contents has been provided (lines 9-30).

What are the unique characteristics of b-HMPA-derived dendrimers and their potential applications in gene therapy and biomedical fields?

The unique characteristics of b-HMPA-derived dendrimers have been reported in lines 20-22, 24-26, 107-1109, 845-854, 1169-1172, 1192-1196, and 1376-1379. Specifically, they are unique, because they can be produced on a large scale at low-cost due to the commercial availability of the AB2 monomer b-HMPA at low price. Additionally, b-HMPA is a very versatile molecule which allows to synthesize several different architectures, which, differently from the well-known PAMAMs and PEIs, are all endowed with a polyester biodegradable scaffold which assure biocompatibility and low cytotoxicity. Moreover, dendritic materials derived from b-HMPA represent a unique class of polymers destined to find increasing uses in various fields, including the biological areas, thanks to the reliability of the synthetic procedures, possibility of post-functionalization and hybridization. Information concerning their application in biomedical fields has been included in paragraph 3.4. It has been reported that a large variety of carefully selected bioactive molecules have been successfully covalently linked to such dendritic structures, including carbohydrates (mannose, sialic acid, β-D-lactopyranose), amino acids (cysteine), peptides (glutathione or GSH), anticancer agents (doxorubicin) and antibiotics (amoxicillin) (lines 1169-1176).
Additional information concerning the possible biomedical applications of b-HMPA dendrimers has been added in lines 1187-1190 and in paragraph 5 in lines 1339-1340.

In the Introduction section, the authors are encouraged to state informative objectives to guide the readers.

Some informative objectives were already present in the original version of the manuscript. Anyway, additional details have been included in the revised version. Please, see lines 109-124.

Figures should be thoroughly polished to improve the overall presentation of the manuscript. Ensure that text in figures is clear and consider making color changes where appropriate.

All Figures and Schemes have been checked and when necessary, they have been improved to make the text clear. Colours of Figure 1 have been changed.

How have b-HMPA-based dendrimers evolved over time, and what is the specific focus of the paper regarding amino acid functionalization?

Information concerning the evolution over time of b-HMPA-based dendritic materials (from simple dendrons to dendronized surfaces) is included in the text where the narration of the historical development of several different materials constructed with b-HMPA or its anhydrides has been reported (Section 3, and specifically lines 896-1167). Regarding the amino acids functionalization of b-HMPA-based dendrimer scaffolds, the specific focus of the paper was to report our achievements, as described in the abstract, in the introduction, in Section 4, Table 4 and Conclusions.

Please include the reaction mechanism in a schematic form to enhance the visual representation of the process.

As asked, reaction Schemes (Schemes 25-28) concerning the preparation and amino acids functionalization of our 1st age G4 and G5 dendrimers have been included.

What is the primary objective of gene therapy and why are nonviral cationic materials, such as dendrimers, considered as alternatives to viral vectors in this context?

The primary objective of gene therapy has been reported and explained both in the abstract (lines 9-13), in the introduction (lines 64-71), and Section 2, while the reason for which cationic materials are considered a good alternative option to the use of viral vectors has been explained in the abstract (lines 13-22), and in the introduction (lines 72-109).

The conclusion should be rewritten to be more concise and provide an overarching summary of the findings and their significance. for example what is the main contribution of the authors to this field, and how does it relate to the broader context of gene therapy and dendrimer-based materials?

As asked, conclusion have been rewritten also highlighting our main contribution to this field, and how does it could relate to the broader context of gene therapy and dendrimer-based materials (lines 1366-1379).

Replace older references with more recent ones to ensure the relevance and currency of the sources.

We have accompanied our work with references as recent as possible, but since the work is a narrative history of the birth and development over time of the various materials derived from b-HMPA, which found their first examples in the 80’, it was necessary to include also several references from those years.

Reviewer 2 Report

Comments and Suggestions for Authors

Comments on the Quality of English Language

Author Response

In this manuscript, a review study is provided for gene therapy and the primary cationic materials suggested for the transportation and delivery of genetic materials. The manuscript is acceptable after these minor changes.

1. Please also describe which other materials can be used for gene therapy in the introduction part.

As asked, a brief description of other materials, except for dendrimers, which can be used for gene therapy has been included in the introduction section (lines 64-71).

2. Figure 1 is not clear.

Figure 1 has been modified to make it clearer.

3. Other review articles on this topic have been already reported. What is the significance of this study, which should be further clarified and emphasized?

We agree with the Reviewer about the existence of other reviews on this topic, but all of them report only on the research work of other authors. On the contrary, given our personal experience in synthesizing b-HMPA-based polyester dendrimers different from those prepared by other authors, we have originally provided information on our production, which could be of interest for other researchers, for further studies. This fact has been emphasized both in the abstract, in the introduction and in the conclusions.

4. The manuscript has some grammatical and typing errors, and it needs further polishing.

All manuscript has been checked to reduce all typos and grammatical errors. Additionally, it was revised by our colleague Prof. Deirdre Kantz, English teacher mother tongue working for the University of Genoa and Pavia, where she teaches scientific English in the degree courses in Pharmacy and Pharmaceutical Chemistry and Technology.

5. For some citations, DOI numbers are missing. All references should be in the same style

The missing DOI numbers have been added. The style of references has been standardized.